# MATE: Plugging in Model Awareness to Task Embedding for Meta Learning

**Xiaohan Chen, Zhangyang Wang**
Department of Electrical and Computer Engineering
University of Texas at Austin
{xiaohan.chen, atlaswang}@utexas.edu

**Siyu Tang**
Department of Computer Science
ETH Zürich
siyu.tang@inf.ethz.ch

**Krikamol Muandet**
Max Planck Institute for Intelligent Systems
Tübingen, Germany
krikamol@tuebingen.mpg.de

## Abstract

Meta-learning improves generalization of machine learning models when faced with previously unseen tasks by leveraging experiences from different, yet related prior tasks. To allow for better generalization, we propose a novel task representation called *model-aware task embedding* (MATE) that incorporates not only the data distributions of different tasks, but also the complexity of the tasks through the models used. The task complexity is taken into account by a novel variant of kernel mean embedding, combined with an instance-adaptive attention mechanism inspired by an SVM-based feature selection algorithm. Together with conditioning layers in deep neural networks, MATE can be easily incorporated into existing meta learners as a plug-and-play module. While MATE is widely applicable to general tasks where the concept of task/environment is involved, we demonstrate its effectiveness in few-shot learning by improving a state-of-the-art model consistently on two benchmarks. Source codes for this paper are available at https://github.com/VITA-Group/MATE.

## 1 Introduction

Human can often quickly learn new concepts after seeing only a few examples or having minutes of experience. Prevalent deep neural networks, on the other hand, require enormous amount of data to learn to generalize well [32]. To avoid overfitting, these gigantic deep learning models must integrate prior knowledge like humans do. For example, we can quickly learn to distinguish between several characters in a brand-new language we have never learned before because we can exploit past experiences and concepts that we have acquired in our native language. In machine learning, computer vision, and robotics, prior experience often comes in the form of *tasks* and their relationships. For example, after teaching a robot to walk, we expect that it should be able to learn to run faster by exploiting the related skills acquired from the previous task. Therefore, it is vital for more efficient meta-learning to find a method that can summarize and make use of the prior experience.

The key of meta-learning, or learning to learn, is to learn an *inductive bias* for the new task using data from previous tasks [3, 46]. In principle, machine learning problems can be viewed as a problem of searching for the best hypothesis in a hypothesis space $\mathcal{F}$ that characterizes the inductive bias. Maximum margin bias for support vector machine (SVM) [52] and minimum feature bias for the feature selection algorithms [8] are examples of how the inductive bias can be incorporated. Furthermore, the *architectures* of deep neural networks used, e.g., the popular 2D-convolution blocks

and residual/dense connections [15, 18], are now increasingly viewed as a form of inductive bias too [51, 16, 9]. Finding the most suitable inductive bias for the problem at hand not only ensures that good solutions can be found by a learning algorithm, but can also expedite the learning.

In general, the structure of the hypothesis space $\mathcal{F}$ determines 1) its capacity; and thus 2) the performance of the *optimal* hypothesis $f^* \in \mathcal{F}$; moreover, 3) the difficulty of identifying $f^*$ in $\mathcal{F}$. On the one hand, the family of deep networks characterizes a gigantic and special hypothesis space which potentially contains good solutions, but is notoriously difficult to train. Training deep networks from scratch on each task with limited amount of data often lead to overfitting. On the other hand, let us consider an extreme case where the hypothesis space only contains one element, i.e., $\mathcal{F} = \{f^*\}$ where $f^*$ denotes the optimal solution. This effectively reduces the sample complexity to zero, i.e., no learning is required. In this sense, the core problem of meta-learning is to construct a suitable hypothesis space that contains an optimal solution and is at the same time easy to learn. Model-Agnostic Meta-Learning (MAML) [10] can be perceived as constraining the hypothesis space to be within the neighborhood of the meta-parameters. Since the optimal parameters for different tasks are assumed to lie in this neighborhood, they can be reached via a few steps of gradient decent.

The above reasoning implies our core idea to advocate in this paper: **the model learned in each task is itself part of the inductive bias**. Knowing which models work best for previous tasks should contribute to improving transferability to new tasks that employ similar models. A natural idea that follows is to extract representations of the tasks, and to establish a relationship between the tasks and their corresponding best models. Most previous works that either implicitly or explicitly leverage task representations only refer to the data distribution when constructing the representation [35, 49], with a recent exception in [1]. However. we conjecture such might not suffice for sufficient inductive bias.

Take few-shot classification, which will be a main application scenario considered by this paper, as an example. In a typical few-shot setting, due to the small class number as well as the small training sample size, it is reasonable to assume a model will make more use of *compact, essential* features to differentiate classes, compared to general classification with abundant training data covering vast variations. For instance, a few-shot classifier might find it sufficient to classify dog and car, by just examining ear and fur features. However, those "essential" features vary with tasks, e.g., the same ear-and-fur feature will become no longer "essential" if learning a few-shot dog-cat classifier. Therefore, in addition to the data distribution, the features that a classifier learns to focus on can contribute complementary information to characterizing the tasks and constructing the embeddings.

## 1.1 Our Contributions

**Framework.** In order to inject the model inductive bias, we propose a *model-aware task embedding* (**MATE**) framework which is generally applicable as a *plug-in* module to most meta-learning (single) models. The proposed representation is based on the *novel* Hilbert space embedding of distributions [43, 29] which can capture information of both data distribution and the model used in learning. Although similar embeddings have been *implicitly* applied to meta-learning [1, 35, 49], our framework is the first to: 1) be *model-aware* via the incorporation of the model information into the embedding, instead of relying only on data distribution; 2) *explicitly* draw a connection between meta learning and Hilbert space embedding of distributions [43, 29] from a theoretical perspective.

**Methodology.** For incorporating model information, we propose an *instance-adaptive soft* feature-selection method inspired by a first-order variable selection method in [37], which adaptively emphasizes essential features that vary per each task. We view it interpretable and generalizable to meta-learning applications even beyond few-shot learning (a main study subject of this paper).

**Experiments.** We demonstrate that MATE can help the learning agent to adapt faster and better to new tasks, thanks to the new model-aware inductive prior guiding to constrain the hypothesis space. We illustrate that this new inductive bias is highly informative and adaptive across tasks, as a result of the proposed instance-adaptive soft feature-selection. We empirically demonstrate on two few-shot learning benchmarks that MATE improve up to 1% 5-shot accuracy, on top of a state-of-the-art meta learner "backbones", showing MATE to be generally effective and easy-to-use.

## 2 Related Works

**Meta-Learning for Few-Shot Learning.** Although this paper focuses specifically on few-shot learning as the main application example, our method can be applied to general meta-learning

scenarios [31, 41, 47]. A popular line of works for meta-learning focuses on learning how to update the learner's model parameters [4, 5, 42]. This approach has been applied to learning to optimize deep neural networks [2, 17, 24] and dynamically changing recurrent neural networks [14]. In [38], the authors suggest to learn both the weight initialization and optimizer for few-shot image recognition.

The next line of works in meta-learning are *metric-based*. Matching Network [53] learns a feature extractor and compares a pair of data using cosine similarity. It is applied to one-shot learning, where every testing sample is compared to the reference sample in each class in the support set, i.e., the training data we can refer to for model adaptation in one task. Prototypical Network [44] extends matching network to few-shot learning by comparing test samples with the *prototypes*, computed as the class-conditional mean embeddings. Relation Network [45] learns the comparison metric instead of using a pre-defined metric such as cosine similarity. Task Dependent Adaptive Metric (TADAM) [35] incorporates more adaptation to improve over [45] during meta-testing by learning a task-dependent metric. Lately, Category Traversal Module (CTM) [23] focuses only on task-relevant features by learning to correlate the prototypes of all classes. Our intuition of making the meta learner *model-aware* echoes that of CTM [23]. However, MATE's methodology grounds its idea on the Hilbert space embedding theory [43, 29], and has interpretability benefits from instance-adaptive soft feature-selection inspired by first-order variable selection [37]. As MATE injects model-aware task representations by conditioning the backbones, while CTM acts as sample-feature post-processing, the two might also be applied together in future work.

The last line of works are *optimization-based* approaches. MAML [10] is arguably one of the best-known along this line. It learns a set of parameters that can be adapted to new tasks easily within a few steps of gradient descent. More scalable variants of MAML include FOMAML [33] and Reptile [34]. Bayesian-MAML (B-MAML) [57] further combines the gradient-based adaptation with variational inference in a probabilistic framework to model more complex uncertainty beyond a simple Gaussian approximation. [40] proposes a Latent Embedding Optimization (LEO) framework that decouples the gradient-based adaptation procedure from the underlying high-dimensional parameter space, and place the former in a lower-dimensional latent space. Lastly, alternative optimization-based methods train a meta feature extractor followed by different base learners such as ridge or logistic regression in [6, 49], and support vector machine (SVM) in [21].

**Task Embedding.** To exploit information about task relationship, Taskonomy [58] explores the structure of the space of tasks, focusing on the scenario of transfer learning in a curated collection of 26 visual tasks. [49] computes pairwise task transfer distances to form a directed hierarchy. [1] proposes *Task2Vec*, a task embedding based on the estimation of the Fisher information matrix associated with parameters in the so-called *probe network* that is used to extract features from images, along with metrics to evaluate task similarity with the proposed task embedding.

Notably, while Task2Vec is solely dependent on the task, [1] extend their Task2Vec to Model2Vec by applying a model-dependent additive correction term to the task embedding, which is optimized to model the interaction between the task and model. However, the optimization for correction terms require *a set of models*, while most recent meta-learning frameworks assume to learn *a single model* [6, 10, 21, 27, 40, 49], limiting the general applicability of [1]. Besides, [1] also did not discuss nor experimentally benchmark on meta-learning tasks such as few-shot classification.

On a separate note, although many meta-learning frameworks [21, 40] have achieved impressive results on few-shot learning benchmarks such as CIFAR-FS and miniImageNet [53], task representation remains to be rarely exploited. This paper is motivated to close this gap: showing the importance of task representation in meta-learning/few-shot classification, both theoretically and experimentally.

**Domain Generalization.** The goal of domain generalization is to generalize models to unseen domains without knowledge about the target distribution [7]. [28] proposes to learn a data transformation that maps data from different domains to a shared low-dimensional subspace and that subspace can be extended to unseen domains during testing. The idea of learning domain-invariant features has also been proposed in [13, 22, 55]. [50] proposes to use MMD loss as a regularization to minimize the distance of kernel mean embeddings of different tasks to learn a domain invariant representation.

## 3    MATE: Model-Aware Task Embeddings

Before diving into technical details of our new task representation, we first give an intuition at a conceptual level as to why model information might improve task representation in meta-learning.

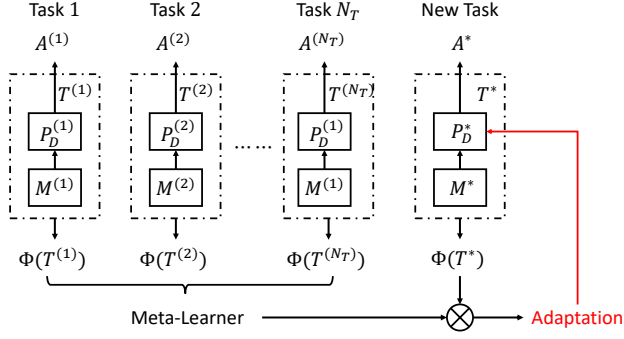

Figure 1: An illustration of model awareness for meta-learning. Each task is represented by a tuple $T^{(l)} = (M^{(l)}, P_D^{(l)}, A^{(l)})$, where model $M^{(l)}$ is applied on data distribution $P_D^{(l)}$ with some performance measure $A^{(l)}$. We learn to construct a joint distribution of the data distribution and the model: $\Phi(T^{(l)}) = \Phi(P_D^{(l)}, M^{(l)})$ to capture the task complexity, and use it to train a meta-learner so that the new meta-learner knows how to adapt w.r.t. a new task $T^* = (P_D^*, M^*)$.

Consider $N_T$ tasks where each task can be represented by a tuple $T^{(l)} = (M^{(l)}, P_D^{(l)}, A^{(l)})$ for $l = 1, \ldots, N_T$ where $M^{(l)}$ denotes the model, $P_D^{(l)}$ denotes the data distribution, and $A^{(l)}$ denotes a task-specific performance measure, e.g., classification accuracy, associated with the model $M^{(l)}$. For each task $i$, we construct two task representations: $\Phi(P_D^{(l)})$ and $\Phi(P_D^{(l)}, M^{(l)})$. The former relies only on the data distribution, whereas the latter also takes the model into account. Provided with the new task $T^* = (M^*, P_D^*)$ where $M^*$ may represent the potential model for that task, we construct the representations $\Phi(P_D^*)$ and $\Phi(P_D^*, M^*)$. Then, it is clear that when deciding which tasks among all $T^{(l)}$ are most relevant to $T^*$ on the basis of $A^{(l)}$, it is impossible for $\Phi(P_D^*)$ and $\Phi(P_D^{(l)})$ to capture aspects of the models $M^*$ and $M^{(l)}$ that might contribute to the task performance. On the other hand, $\Phi(P_D^*, M^*)$ and $\Phi(P_D^{(l)}, M^{(l)})$ can capture this information through the joint representation, as illustrated in Figure 1. Moreover, it allows us to assess the task complexity w.r.t. the model.

To formalize our definition, let $T = (P_D, M)$ denote a task which consists of a data distribution $P_D = P_D(X, Y)$ over some input space $\mathcal{X} \times \mathcal{Y}$ and a model $M$ mapping from $\mathcal{X}$ to an output space $\mathcal{Y}$. We assume that $T$ is distributed according to some unknown distribution $\mathscr{P}(\mathcal{P}_D, \mathcal{M})$ over the product space of distributions $P_D$ and the model $M$. Let $\phi : \mathcal{X} \to \mathcal{F}$ be a feature map of $X$ into the feature space $\mathcal{F}$ and $f_M : \mathcal{X} \to \mathcal{F}$ a model-dependent bounded continuous function such that $f_M(x)_i > 0$ for all index $i$ and $x \in \mathcal{X}$.

Given any task $T = (P_D, M)$, we propose to represent it using the following conformal representation,

$$\Phi(T) := \int f_M(x) \odot \phi(x) \, \mathrm{d}P_D(x). \tag{1}$$

where $\odot$ is an elementwise multiplication. Note that $\Phi(T)$ may also depend on $Y$ through $f_M$. Intuitively, the function $f_M(x)$ acts as a reweighting function of the feature map $\phi(x)$. Here, $\phi$ can be a feature map associated with the kernel function or the last layer of deep neural networks. If $\phi$ corresponds to the canonical feature map of the characteristic kernel $k : \mathcal{X} \times \mathcal{X} \to \mathbb{R}$, i.e., $\phi(x) = k(x, \cdot)$, and $k(x, x)|f(x)|^2$ is bounded, the map defined by (1) is injective, i.e., the representation $\Phi(T)$ captures all information about the task $T$ [11, Lemma 2]. Note that (1) can also be seen as a classical kernel mean embedding with the conformal kernel $\tilde{k}(x, x') = f_M(x)f_M(x')k(x, x')$ [54].

The intuition for (1) is as follows. If $f_M(x) = c \cdot \mathbf{1}$ for some constant $c > 0$ and $\mathbf{1} \in \mathcal{F}$, the all-one vector and $\phi$ in the feature space is the canonical feature map of the kernel $k$, then (1) reduces to a classical Hilbert space embedding of distributions, completely ignoring information about the model $M$. In general cases, we expect $f_M$ to inform the "hardness" of the task with respect to the distribution $P(X)$. Given an i.i.d sample $x_1, \ldots, x_K$, our representation (1) can be estimated by

$$\widehat{\Phi}(T) := \frac{1}{K} \sum_{k=1}^{K} f_M(x_k) \odot \phi(x_k). \tag{2}$$

The above representation, **when $f_M$ is not present**, is commonly used in domain adaptation [59, 25], domain generalization [13, 22, 28, 50], and meta-learning [1, 35, 49]. Hence, the **key novelty** of (2) is on incorporating the model-dependent function $f_M$ into the representation.

## 3.1 Model-Aware Surrogate Functions

Recall that for all coordinates $i$ and $x \in \mathcal{X}$ for (1) to be well-defined, we only require

$$f_M(x)_i > 0 \quad \text{and} \quad \langle f_M(x) \odot \phi(x), f_M(x) \odot \phi(x) \rangle < \infty.$$

Moreover, all information about the task is preserved if the kernel defined by the feature map $\phi$ is characteristic. Here, we present our design of the $f_M$ function to utilize the model information, which otherwise cannot be captured solely using data distribution.

In a general framework of few-shot classification, a classifier works on top of the feature extractor $\phi$. The classifier could be either optimization-based (e.g. SVM in [21]) or metric-based [44, 23]. Given a new task, the classifier is trained using features extracted from a small *support set* $D_s$: $\{\phi(x)\}_{x \in D_s}$. Then, we apply the trained classifier to predicting on the *query set* $D_q$. However, we argue that not all features in $\phi(x)$ are *essential* to one specific task, especially in few-shot classification where there are only a very small number of classes and each class usually has less than 10 training samples. Moreover, the *essential* features can vary significantly across different tasks. In constrast, the task representation generated by the conventional Hilbert space embedding with $f_M = c \cdot \mathbf{1}$ cannot capture these variations due to absence of the classifiers' own information.

Inspired by [21] and [37], we propose to add an independent SVM classifier parameterized by $\omega$ on top of the feature extractor. We use the $D_s$: samples to solve the optimal $\omega^*$, instead of the original classifier. In order to incorporate the model information, we consider the following specific $f_M$:

$$f_M(x) = c \cdot \left| \frac{\partial \|\omega^*\|_2^2}{\partial \phi(x)} \right|, \tag{3}$$

where $c$ is a hyperparameter. In SVM, $\|\omega^*\|_2^2$ is the optimal objective value, e.g., the optimal margin. The $f_M$ function defined in (3) then selects the dimensions that will account for the most significant changes in the margin, if slightly perturbed. Ideally, the dimensions that are orthogonal to the margin will be select as essential features because changes on those dimensions will affect the margin most. Similar ideas to this SVM criteria were exploited by first-order methods in variable selection based [37]. Our method can also be seen as an instance-adaptive *soft* feature selection or attention.

## 3.2 Sample-Task Feature Fusion

Assume that we extract both the sample feature $\phi(x_i)$ and the task feature $\hat{\Phi}(T)$, where $\hat{\Phi}(T)$ is constructed using (3). The most naive fusion approach is to concatenate them (CAT):

$$\psi_{cat}(x_i, T) := concat(\phi(x_i), \hat{\Phi}(T)). \tag{4}$$

However, this is very limited because it simply gives the same translation to all samples.

In this work, we use the task representation $\hat{\Phi}(T)$ to modulate a part of the feature extractor $\phi$, similarly to TADAM [35]. Our implementation adopts the **FiLM conditioning layer** [36]. Specifically, A FiLM layer is defined as $\mathrm{FiLM}(\boldsymbol{x}) = \boldsymbol{\gamma} \odot \boldsymbol{x} + \boldsymbol{\beta}$, where $\boldsymbol{\gamma} = \boldsymbol{\gamma}(\hat{\Phi}(T))$ and $\boldsymbol{\beta} = \boldsymbol{\beta}(\hat{\Phi}(T))$ are the scaling and shifting parameters, that are both generated from $\hat{\Phi}(T)$ through a small MLP network, In practice, we place a FiLM layer after every batch normalization (BN) layer of a deep feature extractor.

Note that the learning of $\hat{\Phi}(T)$ and $\phi$ are dependent on each other. To *initialize* $\hat{\Phi}(T)$ at the first place, we first extract task-independent samples features $\phi(x)$ with default task representation, i.e. all-zero task representation which yields identity FiLM layers. Then we construct the model-aware task representation $\hat{\Phi}(T)$ as described in Sec. 3.1. At last, this task representation is used to update the FilM layer and the feature extractor jointly. We also note that the injection of task representation $\hat{\Phi}(T)$ will cause a shift of statistics in the intermediate activations in $\phi$ and thus fail the BN layers during validation or testing. Similar to [56], we craft two sets of BN layers (dual BN), one for task-independent feature extraction with default task representation, and the other for task-dependent feature extraction with runtime model-aware task representation $\hat{\Phi}(T)$.

We adopt the episodic formulation in [53] for the $N$-way, $K$-shot task definition, where each *episode* $D^{(l)}$ is a set of data samples independently drawn from the data distribution $P_D^{(l)}$. Episodes in a dataset are divided into three sets — *meta-training* $\mathcal{S}_{\mathrm{trn}}$, *meta-validation* $\mathcal{S}_{\mathrm{val}}$, and *meta-testing* $\mathcal{S}_{\mathrm{tst}}$. The meta-training and meta-validation sets are accessible during the *meta-training* phase, and the trained model will be tested on the meta-testing set with unseen tasks over the same $\mathcal{P}_D$ In

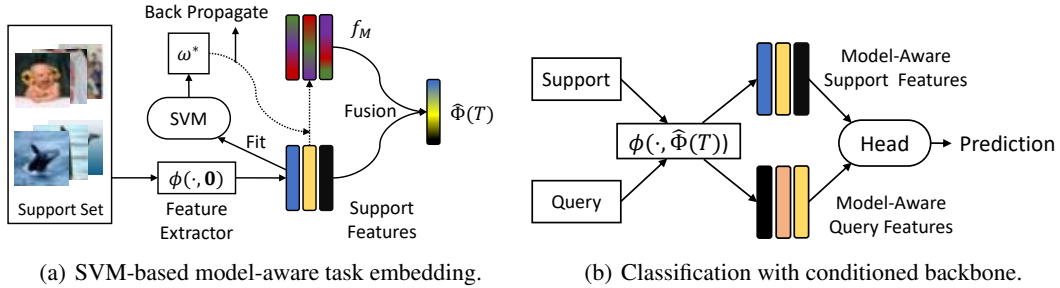

| (a) SVM-based model-aware task embedding. | (b) Classification with conditioned backbone. |

Figure 2: **Overview of the framework.** MATE can be applied on top of general few-shot learning framework with single feature extractor as a plugin module. (a) An SVM classifier is first appended after the backbone $\phi$ and trained with the task-agnostic sample features. Then a set of weights $f_M$ is produced by the proposed instance-adaptive soft feature-selection (3), which are then used to fuse all sample features into the model-aware task embedding $\hat{\Phi}(T)$. (b) the model-aware task embedding is fed to condition the FiLM layers in the backbone feature extractor as adaptation to the current task.

each episode, the task consists of a support (or training) set and a query (or validation) set, i.e., $D^{(l)} = \{D_s^{(l)}, D_q^{(l)}\}$. The model is trained to adapt to this specific task with data from $D_s^{(l)}$. $D_q^{(l)}$ is used to evaluate the model's generalization ability.

The proposed method can be applied on top of most few-shot learning methods that have a "backbone" feature extractor. As shown in Figure 2, a base model consists of a backbone feature extractor $\phi$ and a *classification head* on top of it. This classification head could be comparison-based, optimization-based e.g. a logistic regression or SVM classifier, or a small neural network.

To apply MATE, we first insert FiLM layers, which are conditioned by the representation of current task $\hat{\Phi}(T)$, after batchnorm layers in the backbone $\phi$, so that the backbone now takes two inputs $\phi(x, \hat{\Phi}(T))$. Then we run two passes through the extended backbone. In the first run, we use the default task representation i.e. the all-zero vector that yields identity FiLM layers, meaning without considering the representation of the task. These *task-unaware* features are used to train an SVM classifier as in [21] and summarizes the task representation $\hat{\Phi}(T)$ as shown in section 3.1. Then a second pass of $\phi$ is run with $\hat{\Phi}(T)$ conditioning the FiLM layers. The whole framework is meta-trained with loss function $\mathcal{L}$. We will keep the loss function used by the base model as is, the specific form of which depends on the choice of classification head in the base model. As MATE only makes small changes to the base model and keep all other settings unchanged, it can be easily applied to most of few-shot learning frameworks in a plug-in manner.

**Dual batch normalization.** The backbone $\phi$ will be run for two passes, one of which involves FiLM conditioned on the task representation. However, the conditioned FiLM can change the statistics of intermediate features, such that batchnorm layers cannot estimate stably. We hence adopt the dual batch normalization, originally proposed in [56] to separate clean and adversarial image statistics. Here we use it for a new purpose of separating model-agnostic and model-aware feature statistics.

## 4 Experiments

In this section, we first describe the implementation details (Section 4.1), and then benchmark MATE on two few-shot classification datasets, CIFAR-FS [6] and miniImageNet [53] (Sections 4.2 and 4.3). We complement our quantitative results with a visualization of the embedding produced by MATE compared to model-agnostic embeddings (see **Supplementary**). The implementation of this paper is adapted from the official implementations of MetaOptNet[1].

### 4.1 Implementation Details

**Backbone feature extractor $\phi$.** We mainly apply MATE on top of MetaOptNet [21], due to its state-of-the-art few-shot learning performance, which we aim to improve further. It uses a ResNet-12 [15] backbone as feature extractor. We faithfully follow the setting and regularization tricks such as DropBlock [12], to avoid the overfitting risk. The output of the last residual layer is the sample feature (without applying global averaging pooling). More details of $\phi$ are presented in **Supplementary**.

**Encoding FiLM layers.** FiLM layers are inserted into the backbone $\phi$ after each batchnorm layer and the scaling and shifting parameters $\gamma$ and $\beta$ are encoded from the task representation $\hat{\Phi}(T)$ extracted according to (2) using a two-layer MLP network with LeakyReLU activation (with negative slope 0.1) after each layer. The parameters $\gamma$ and $\beta$ have the same dimensions as the channel numbers of the input feature maps, and are applied to the input features channel-wise. FiLM layers are encoded with independent MLP networks with no weight sharing.

**Optimizer.** We identically follow the practice in [21] for fair comparison. We use stochastic gradient descent (SGD) with momentum 0.9 and weight decay 0.0005. The SGD starts with an initial learning rate of 0.1, that is decayed to 0.006 at epoch 20. The meta-training phase takes a total of 30 epochs, each epoch consisting of 1,000 mini-batches. Each mini-batch samples 8 episodes.

**Episode Setting.** In each episode, we fix the number of classes at 5 (5-way), for both meta-training and meta-testing phases, on both datasets. The query set in each episode contains 6 samples per class during meta-training phase, and 15 samples per class for meta-testing. It was found in [21] that using more shots in meta-training than in meta-testing (1- or 5-shot) leads to better accuracies. We follow this practice to use 15-shot episodes for miniImageNet and 5-shot for CIFAR-FS. On both datasets, We use a total number of 1,000 episodes during meta-testing and report the mean accuracy and its standard deviation over all episodes.

**Other Regularizations.** We follow [21] identically in using regularization technique used during meta-training, such as data augmentation, label smoothing (on miniImageNet), and early stopping. The early stopping is applied at two levels: 1) we only compute 3 iterations when training SVM during meta-testing; 2) we validate the model after each epoch, on 2,000 mini-batches of 5-way 5-shot episodes sampled from the meta-validation set, to terminate training once the validation accuracy starts to drop. We also leverage a regularization technique from [26] to diversify the FiLM parameters $\gamma, \beta$, avoiding collapse in training. We will introduce more details in the **Supplementary**.

### 4.2 Experiments on CIFAR-FS

**CIFAR-FS** [6] is a popular few-shot classification benchmark, designed to be more complicated than the previous Omniglot [20] yet more compact than miniImageNet [53]. It is a variant of CIFAR-100 [19], by randomly splitting it into meta-training, meta-validation and meta-testing sets, containing 64, 16 and 20 classes, respectively. Each class in CIFAR-FS contains 600 images of size $32 \times 32$.

**Comparison Methods.** The results of 5-way classification on CIFAR-FS are shown in Table 1. We compare against a few competitive methods, including MAML [10], ProtoNets [44], Relation Networks [45], R2D2 [6], MetaOptNet [21] and RFS [48]. In [21, 48], the authors report on two settings: the former trained using the combined set of the meta-train and meta-validation set, while the latter using only the meta-train set (hence the former would have better accuracy numbers on the meta-test test due to seeing more data). All results reported in this paper follow the latter setting.

Table 1: Benchmarking results on the CIFAR-FS, compared with previous state-of-the-art works. $\diamond$ Results reported by the original paper: using the the meta-train set only for the meta-training.

| Model | Backbone | CIFAR-FS [6] | |
| --- | --- | --- | --- |
| | | 5-way 1-shot | 5-way 5-shot |
| MAML$^\diamond$ [10] | 32-32-32-32 | $58.9 \pm 1.9\%$ | $71.5 \pm 1.0\%$ |
| Relation Networks$^\diamond$ [45] | 64-96-128-256 | $55.0 \pm 1.0\%$ | $69.3 \pm 0.8\%$ |
| ProtoNets$^\diamond$ [44] | 64-64-64-64 | $55.5 \pm 0.7\%$ | $72.0 \pm 0.6\%$ |
| ProtoNets [44] | ResNet-12 | $71.35 \pm 0.73\%$ | $84.07 \pm 0.51\%$ |
| MATE + ProtoNets | ResNet-12 | $71.49 \pm 0.70\%$ | $84.71 \pm 0.50\%$ |
| R2D2$^\diamond$ [6] | 96-192-384-512 | $65.3 \pm 0.2\%$ | $79.4 \pm 0.1\%$ |
| R2D2 [6] | ResNet-12 | $72.51 \pm 0.72\%$ | $84.60 \pm 0.50\%$ |
| MATE + R2D2 | ResNet-12 | $72.59 \pm 0.0.70\%$ | $85.04 \pm 0.50\%$ |
| MetaOptNet$^\diamond$ [21] | ResNet-12 | $72.0 \pm 0.7\%$ | $84.2 \pm 0.5\%$ |
| MATE + MetaOptNet | ResNet-12 | $72.3 \pm 0.7\%$ | $85.2 \pm 0.4\%$ |

**Result analysis.** Compared with MetaOptNet, "MATE+MetaOptNet" achieves slightly better 1-shot accuracy (72.32% v.s. 72.0%), and much better 5-shot accuracy with an obvious gap (**+1.00%**) The 'MATE+MetaOptNet' model reported in Table 1 uses its MetaOptNet counterpart to initialize the

weights, which makes sense because MATE is used as a plug-in method to improve the base model. We will discuss this in an ablation study to be presented later.

In addition to comparing with ProtoNets [44] and R2D2 [6] on their original small backbones, we also compare with these two methods with larger convolutional backbones. Interestingly, We find that once we try replace the backbone feature extractor with the same ResNet-12 used in MetaOptNet, ProtoNets and R2D2 both show competitive results, and especially R2D2 already performs better than MetaOptNet just by ensuring a fair backbone. Then, MATE can still consistently provide improvements to both (enhanced) baselines: 1) applying MATE to ProtoNets+ResNet12 yields +0.64% 5-shot accuracy and slightly better 1-shot accuracy (+0.14%); 2) applying MATE to R2D2+ResNet12 yields +0.44% 5-shot accuracy improvement and similar 1-shot accuracy (+0.08%). These results also demonstrate that MATE consistently brings more notable gains to 5-shot accuracy than to 1-shot, which is reasonable because we can obtain more accurate information about data distribution on the task with more data and thus task representation of higher quality.

### 4.3 Experiments on miniImageNet

**miniImageNet** [53] is a larger benchmark in which 100 classes are selected from ImageNet [39] and each contains 600 RGB images. (downsampled to 84×84). We follow the popular split in [38] to employ 64 classes for meta-training, 16 for meta-validation and 20 for meta-testing.

We compare with a variety of recent methods, especially those that report on using ResNet-12 back-bones, for fair comparison[2] (see more in the **Supplementary**). To avoid any confusion, MetaOptNet[†] in Table 2 are the results that we replicate to the best efforts with exactly the same setup[3]. After active investigation on this mismatch and intensive communication with the authors of [21], we think it appropriate to report both numbers. In view of the situation, **we humbly suggest that comparing our MATE against MetaOptNet[†] may help draw more fair and meaningful observations.**

Due to much higher dimensionality of the features extracted from miniImageNet samples, we apply one linear layer to the task representation to reduce the dimension before it is used to condition the FiLM layers. Interestingly, we find that freezing this linear layer to its random initialization helps stabilize the training, potentially avoiding overfitting. Results of 5-way classification on miniImageNet are displayed in Table 2. Compared to the best MetaOptNet results that we've replicated, MATE can improve the 1-shot accuracy for 0.44% and 5-shot accuracy for 0.76%.

Table 2: Benchmarking results on the miniImageNet, compared with previous state-of-the-art works.
◇ Results reported by the original paper: using the meta-train set only for the meta-training.
† Results replicated by us to the best effort, by strictly following the official descriptions.

| Model | Backbone | miniImageNet [53] | |
| --- | --- | --- | --- |
| | | 5-way 1-shot | 5-way 5-shot |
| Matching Networks◇ [53] | 64-64-64-64 | 43.56 ± 0.84% | 55.31 ± 0.73% |
| MAML◇ [10] | 32-32-32-32 | 48.70 ± 1.84% | 63.11 ± 0.92% |
| ProtoNets◇ [44] | 64-64-64-64 | 49.42 ± 0.78% | 68.20 ± 0.66% |
| Relation Networks◇ [45] | 64-96-128-256 | 50.44 ± 0.82% | 65.32 ± 0.70% |
| R2D2◇ [6] | 96-192-384-512 | 51.20 ± 0.60% | 68.8 ± 0.10% |
| LEO◇ [40] | WRN-28-10 | 61.76 ± 0.08% | 77.59 ± 0.12% |
| SNAIL◇ [27] | ResNet-12 | 55.71 ± 0.99% | 68.88 ± 0.92% |
| AdaResNet◇ [30] | ResNet-12 | 56.88 ± 0.62% | 71.94 ± 0.57% |
| TADAM◇ [35] | ResNet-12 | 58.50 ± 0.30% | 76.70 ± 0.30% |
| MetaOptNet◇ [21] | ResNet-12 | 62.64 ± 0.61% | 78.63 ± 0.46% |
| MetaOptNet† [21] | ResNet-12 | 61.64 ± 0.60% | 77.88 ± 0.46% |
| MATE + MetaOptNet | ResNet-12 | 62.08 ± 0.64% | 78.64 ± 0.46% |

### 4.4 Ablation study

We perform ablation study on MATE applied to MetaOptNet [21] to investigate the key components of MATE method, from the following perspectives:

- **Sample-task feature fusion.** We compare two ways of feature fusion mentioned in Section 3.2: concatenation (CAT) and FiLM conditioning. Comparing "CAT+KME" and "FiLM+KME", FiLM yields slightly better 1-shot accuracy and we also observe that FiLM induces larger feature variations in visualization. Here KME means kernel mean embedding, i.e. (2) with $f_M \equiv 1$.
- **SVM-based feature attention.** On top of FiLM conditioning, we compare naive KME- and SVM-based task embedding. Combined with SVM-based feature attention with FiLM, we see obvious improvements in 1-shot and 5-shot accuracies (**+0.6%** and **+0.35%** respectively).
- **Other regularizations.** We investigate several techniques that we adopt to avoid training overfitting or collapse. In : *Load backbone* means using a pre-trained backbone of the base model (MetaOptNet) as the initialization; otherwise we use random initialization. *Fix backbone* means only training the MLPs that conditions FiLM layers and the model-aware branch of the DualBN, while freezing the loaded backbone. *FiLM regularization* refers to the one from [26]. We see that *load backbone* boosts 5-shot accuracy but also degrades the 1-shot one. Both *fix backbone* and *regularization*t on FiLM can recover the 1-shot accuracy but *FiLM regularization* improve the 5-shot accuracy remarkably more. Therefore, the last row configuration becomes our default one.

Table 3: Ablation study on MATE + MetaOptNet [21] to investigate key components of MATE.

| Cat | FiLM | KME | SVM | Load Backbone | Fix Backbone | FiLM Regularization | 1-shot | 5-shot |
|---|---|---|---|---|---|---|---|---|
| ✓ | | ✓ | | | | | 71.41% | 84.40% |
| | ✓ | ✓ | | | | | 71.55% | 84.37% |
| | ✓ | | ✓ | | | | 72.15% | 84.72% |
| | ✓ | | ✓ | ✓ | | | 72.01% | 85.13% |
| | ✓ | | ✓ | ✓ | ✓ | | **72.57%** | 84.76% |
| | ✓ | | ✓ | ✓ | | ✓ | 72.32% | **85.20%** |

## 5  Conclusion

This work introduces the model-aware task embedding (MATE), a novel representation that is able to efficiently fuse data distribution and model inductive bias. Built on the Hilbert space embedding of distributions, MATE introduces a model-dependent surrogate function to improve the current kernel mean embedding, that can be incorporated into deep neural networks. We empirically show the general effectiveness of MATE in two few-shot learning benchmarks. Our future work will integrate MATE with more state-of-the-art meta learning models besides [21].

## Broader Impact

Nowadays machine learning models requires training with a large number of samples. Humans, in contrast, learn new concepts and skills much faster and more efficiently. Even a kid who has seen cats and birds only a few times can quickly tell them apart. Inspired by that, meat learning aims to design a machine learning model with similar properties — learning a new task fast over a few training examples, via experiencing and summarizing generalizable rules from a family of similar tasks. Meta learning is significant in at least two-folds: 1) to reduce the labeled samples needed by training models to resolve certain task(s); and (2) to achieve open-end lifelong learning for a stream of tasks by transferring the acquired knowledge.

Our MATE framework leverages a Hilbert space embedding framework and inject model awareness to (in principle) any meta learning algorithm. It shows to help the learning agent to adapt faster and better to new tasks, thanks to this new model-based inductive prior.

## Acknowledgments and Disclosure of Funding

This project is funded through the MPI-IS Grassroots Project 2019: Kernel Methods Meet Deep Neural Network.

## Footnotes

[1]`https://github.com/kjunelee/MetaOptNet`

[2]We did not include a few other methods using deeper backbones, e.g. [23] on ResNet-19, since we find that: 1) MATE on ResNet-19 is prone to overfitting miniImageNet if not carefully tuned, while it is already competitive with ResNet-12; and 2) re-implementing [23] on ResNet-12 catastrophically drops its performance.

[3]We exactly reproduced MetaOptNet on CIFAR-FS, but were unable to close the gap on miniImageNet. The only remaining differences lie in the software environment, and random seeds, which we cannot control.

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
