[Supplementary Material]

# Supplementary Material for
# MATE: Plugging in Model Awareness to Task Embedding for Meta Learning

## 1 Implementation details

### 1.1 ResNet-12 backbone

We use as the backbone a ResNet-12 network, along with DropBlock ([11] in the main text) which is a structured form of dropout serving as a regularization technique to alleviate overfitting.

An illustration of the building block in ResNet-12 `BasicBlock(C,r,b)` is shown in Fig. 1 in this supplementary material, which consists of 1x1 or 3x3 `Conv(C)`, where `C` stands for the number of output channels, batch normalization layers, LeakyReLU activations with 0.1 negative slope coefficient, 2x2 maxpooling layers and `DropBlock(r,b)` layers with `r` dropping rate and `b` block size. Note that when `b` equals one `DropBlock(r,1)` reduces to the normal dropout layer with dropping rate `r`.

On the CIFAR-FS dataset, the backbone consists of four basic building blocks:
`BasicBlock(64,0.1,1)`
`BasicBlock(160,0.1,1)`
`BasicBlock(320,0.1,2)`
`BasicBlock(640,0.1,2)`,
where the DropBlock layers are only applied in the last two building blocks.

On the miniImageNet dataset, we need larger block size for DropBlock because the sizes of feature maps at intermediate layers are larger than those on the CIFAR-FS dataset due to larger input image size (84x84). Therefore, the backbone used for miniImageNet dataset consists of:
`BasicBlock(64,0.1,1)`
`BasicBlock(160,0.1,1)`
`BasicBlock(320,0.1,5)`
`BasicBlock(640,0.1,5)`.

On both datasets, the output of the last block is used as the sample feature (without applying global average pooling).

### 1.2 Regularization on FiLM layers

To regularize the FiLM conditioning layers to prevent them from collapsing, we also leverage a regularization technique from (see ref. [25] in the main text) to diversify the FiLM parameters $\gamma, \beta$, avoiding collapse in training.

Suppose that $[\gamma, \beta] = g(\hat{\Phi}(T))$, i.e. the affine transformation FiLM layers are conditioned on the embedding of the current task $\hat{\Phi}(T)$, where $g(\cdot)$ is a two-layer MLP with leakyReLU activation function in between. The MSGAN regularization term is defined as

DropBlock($r,b$)

BasicBlock($C$, $r$, $b$)    MaxPooling(2)
LeakyReLU(0.1)

Residual Branch    BatchNorm
3x3 Conv($C$)

BatchNorm    LeakyReLU(0.1)
1x1 Conv($C$)    BatchNorm
3x3 Conv($C$)

LeakyReLU(0.1)
BatchNorm
3x3 Conv($C$)

$x$

Figure 1: The basic building block in the ResNet-12 backbone feature extractor.

$$\mathcal{L}_{MS} = \max_{g} \frac{d_F\Big(g(\hat{\Phi}(T_1)), g(\hat{\Phi}(T_2))\Big)}{d_T(\hat{\Phi}(T_1), \hat{\Phi}(T_2))},  \tag{1}$$

where $d_F$ and $d_T$ are similarity metrics in the representation space of parameters of FiLM affine transformation and the task embedding space, respectively. In this paper, we take $d_F$ and $d_T$ as the $\ell_1$ norm. $\mathcal{L}_{MS}$ is reweighted with coefficient $\lambda_{MS}$ and added to the main loss function used by the base model. $\lambda_{MS}$ is set to $10^{-6}$ in this paper.

### 1.3   Early stopping

It is found on both miniImageNet and CIFAR-FS datasets that the models will quickly suffer from overfitting with a few epochs of training after the first learning rate decay (at the end of the $20$-$th$ epoch). Therefore, during model selection, we will record the first 5 models after the first learning decay, i.e. models after epoch 21 to 25, and select the model with highest validation accuracy.

## 2   Visualization results

We show visualization results on CIFAR-FS dataset. We mainly compare three models: the MetaOpt-Net (see [20] in the main text) baseline, KME task embedding (model-agnostic) with FiLM conditioning and SVM-based model-aware task embedding with FiLM conditioning, i.e. the row 2 and row 3 in Table 3 in the main text. For fair comparison between the latter two, we do not apply the regularizations in Table 3 any of the two models.

We randomly sample 10 episodes, i.e. 10 tasks from the meta-testing set of CIFAR-FS under the 5-way 5-shot setting, and then extract the features before the final classifier on top. The randomly sampled tasks contains 18 different classes. We then embed all the samples features into 2-D space using t-SNE. The results are shown in Figure 2 below. We can see from the visualization results that, the model-agnostic KME task embedding is inefficient and can even make the sample feature embeddings look worse. This might help to understand why the combination of KME and FiLM

conditioning causes a performance degradation on the 1-shot tasks. In contrast, the proposed model-aware SVM-based task embedding, when co-working with FiLM conditioning, forcing the samples features to be clustered better and more separable.

(a) MetaOptNet        (b) KME + FiLM        (c) SVM + FiLM (MATE)

Figure 2: Visualization of feature samples using (a) baseline MetaOptNet, (b) KME (model-agnostic) task embedding + FiLM conditioning and (c) SVM-based model-aware task embedding + FiLM conditioning, i.e. MATE.

## 3 Comparison with previous works with re-implemented comparable backbone

As we discuss in the footnote in section 4.3, we did not include a few using deeper backbones, e.g. CTM ([22] in the main text) on ResNet-19, since we find that: 1) MATE on ResNet-19 is prone to overfitting miniImageNet if not carefully tuned, while it is already competitive with ResNet-12; and 2) re-implementing CTM on ResNet-12 catastrophically drops its performance (from 80.51% to 59.52% on 5-way 5-shot cases).

We use the officially released implementation of CTM and try our best to tune the parameters. However, as other followers find, it seems hard to reproduce the reported results in [22]. In-place replacement of the ResNet-19 backbone as is used in [22] with the same ResNet-12 as we use in this paper will further degrade the performance of CTM.