[Reviews · NeurIPS 2020]

Review 1

Summary and Contributions: The work advocates to bring in model awareness to learning task embeddings for meta learning/few-shot classification. The main argument is that existing works only leverage the data distribution, while the models used in learning the embeddings can provide additional value by encoding some measure of task difficulty. The task complexity is taken into account by a Hilbert kernel mean embedding (KME) framework, and the idea is further implemented by an SVM-based feature selection algorithm and FiLM conditioning layers.

Strengths: The proposed idea is well motivated and interesting. “The model learned in each task is itself part of the inductive bias” is a convincing argument; It makes sense that reasoning the difficulty of different tasks by their best-fit models can inform how to leverage those tasks to generalize to new tasks. The authors clearly layout this main idea in Section 3 beginning using KME theory. Section 3 then did a good job in connected their theory to concrete implementations such as the model-aware surrogate using SVM variable selection, and the sample-task fusion by DNN conditioning layers. This method is general and in principle should be appliable to many few-shot backbones. In experiments, the authors plugged in MATE on top of the current state-of-the-art in few-shot classification (MetaOptNet, reimplemented by authors), and reported two benchmarks. This “plug in” model awareness improves CIFAR-FS top-5 by a noteworthy 1%, and miniImageNet top-5 by 0.76%. Those look promising considering MATE essentially used the same MetaOptNet model in the implementation except for an extra model-aware conditioning step at the task embedding stage. The ablation study dissects the respective gain of each MATE component, which is also helpful.

Weaknesses: One clear limitation is that authors demonstrated MATE idea on only the MetaOPTNet model, despite their more general claim and design. Perhaps this could be partially excused, as the authors reported (in supplementary section 3) some unsuccessful trials in reproducing another SOTA method. Also it is kinda known by the field that reproducing meta learning results could be very unstable and challenging. But, this paper could be definitely strengthened if the authors can verify their ideas on more backbones. Another limitation is that the reported performance gain from adding MATE is not very significant (although relatively consistent). While Top-5 demonstrate observable margins, the Top-1 is usually within the confidence interval during the ablation. BTW, it is unclear to me what protocol the authors followed to compute the performance std (e.g. how many runs)? And in Table 2, the last three rows’ top-5 std values are all 0.46% - just to make sure those were not typos? Despite the above, I think the model-aware idea would be of interest and inspiration to the meta learning community, so I currently lean toward acceptance.

Correctness: Yes.

Clarity: The paper is clearly written. The motivation and theory ground are explained well. Figure 2 helps understand the framework

Relation to Prior Work: The authors have well discussed related works.

Reproducibility: Yes

Additional Feedback:


Review 2

Summary and Contributions: This paper argues that, in a meta-learning setup, the learner can not only benefit from incorporating the task distribution but also from representing the model complexity. To this end, the authors propose a model-aware task embedding which is inspired by kernel theory. Specifically, they propose to train a SVM classifier on top of the backbone feature extractor. From the SVM parameter w*, they define the model complexity feature vector as f_M(x) = \partial || w ||^2_2 / \partial \phi(x) They then use this vector to condition the film parameters of the feature extractor in a second pass through this model. Note that in the first pass they set the FiLM parameters to the identity transformation. The entire meta-learning model is then trained in the usual end-to-end way. While the proposed method can be incorporated into any meta-learner with a backbone feature extractor, they experiment with MetaOptNet model (a state-of-the-art model). On CIFAR-FS and MiniImageNet, the authors report small gains over the baseline models.

Strengths: - The introduction of the problem is well-motivated - The proposed method is novel and interesting, and generally applicable to many meta-learning models - The paper is well-written and extensively covers related work

Weaknesses: - My main concern is the experimental section. The current results show only very small benefits over the baseline model. The paper would therefore be much stronger if they would report results for other models than MetaOptNet. This will emphasize the generality of the method and also convincingly demonstrate that the proposed method can improve several baseline models. - Perhaps more a question than a comment: why can’t you condition the FiLM parameters on the w parameter directly? What are the benefits of using your notion? I'm willing to update my score if you can address these concerns.

Correctness: Yes, as far as I can judge

Clarity: Yes, in general the paper was easy to follow.

Relation to Prior Work: Yes, the related work section covers this well

Reproducibility: Yes

Additional Feedback:


Review 3

Summary and Contributions: The paper proposes to improve the generalization to unseen tasks in meta learning, by inventing a new model-aware task embedding framework called MATE. The main new argument is that the model learned in each task is itself part of the inductive bias that could imply the task complexity. The authors derived their framework from kernel mean embedding, and implemented that with instance-adaptive attention plus model conditioning.

Strengths: - The core idea of informing task difficulty by model awareness is interesting and well-illustrated - The authors explicitly draw a connection between meta learning and Hilbert space embedding of distributions from a theoretical perspective. - The idea of incorporating model awareness is implemented with an SVM-style first-order variable selection mechanism and a neural network conditioning layer. The MATE module seems easy to plug-in. - Experiments demonstrate that MATE can help the learning agent to adapt faster and better to new tasks, improving up to 1% 5-shot accuracy on two popular few-shot classification benchmarks.

Weaknesses: - Experiments are not very sufficient to support the claim. While the MATE idea seems to be general, all experiments are done on top of only one recent baseline (MetaOPTNet).

Correctness: Yes

Clarity: Yes

Relation to Prior Work: Yes

Reproducibility: Yes

Additional Feedback:

[Author Response · NeurIPS 2020]

We thank all reviewers for the thoughtful comments and constructive suggestions to improve our paper. In general, all
reviewers find our general message: "*the model learned in each task is itself part of the inductive bias*" convincing.
The core idea of incorporating model complexity into task embedding is "*well motivated and interesting*" (R4), "*novel*
*and interesting, and generally applicable to many meta-learning models*" (R5), and "*is interesting and well-illustrated*"
(R6). The only reservation shared by all reviewers is that experiments are not sufficient to support this claim.

Here, we address the major concern raised by all three reviewers — **generalization to more baselines**. We conducted
additional experiments on two competitive baselines with large backbone feature extractors. To summarize, MATE
brings consistent improvements by exploiting model information in task representations, which confirms our original
finding. We plan to try more baselines and report in the final version. We also provide details about **meta-testing**
**protocol** (R4), discuss **the gain brought by MATE** (R4, R5) and **the choice of FiLM layer conditioning** (R5).

▷ **Applying MATE to more baselines**
**(R4, R5, R6).** Per all your suggestions,
we conducted experiments on two more
baselines. Due to the limited rebuttal time
window and the well-known difficulty in
finding suitable, reproducible implemen-
tation of SOTA meta learning works, we
turn to two baselines that have been com-
pared in this paper, namely, Prototypical

| Model | Backbone | 5-way 1-shot | 5-way 5-shot |
|---|---|---|---|
| MetaOptNet [20] | ResNet-12 | $72.00 \pm 0.70\%$ | $84.20 \pm 0.50\%$ |
| MetaOptNet + MATE | ResNet-12 | $72.30 \pm 0.70\%$ | $85.20 \pm 0.40\%$ |
| ProtoNets [43] | ResNet-12 | $71.35 \pm 0.73\%$ | $84.07 \pm 0.51\%$ |
| ProtoNets + MATE | ResNet-12 | $71.49 \pm 0.70\%$ | $84.71 \pm 0.50\%$ |
| R2D2 [6] | ResNet-12 | $72.51 \pm 0.72\%$ | $84.60 \pm 0.50\%$ |
| R2D2 + MATE | ResNet-12 | $72.59 \pm 0.70\%$ | $85.04 \pm 0.50\%$ |

Networks [43] and R2D2 [6], but use larger convolutional backbones. We limit the experiments on CIFAR-FS, and
will include miniImageNet results on the new baselines. The results are shown in the above table. Although ProtoNets
and R2D2 are somehow old, we would still like to justify that comparing on these two are meaningful and can help to
corroborate the generality of MATE framework. It is known that the original ProtoNets and R2D2 have much lower
performance than more recent works, e.g. they are 12.2% and 4.8% lower in 5-way 5-shot accuracy on CIFAR-FS
compared to MetaOptNet [20], respectively. However, once we try replace the backbone feature extractor with the
same ResNet-12 used in MetaOptNet, ProtoNets and R2D2 both show competative results, and especially R2D2
already performs better than MetaOptNet just by ensuring a fair backbone. Then, MATE can still consistently provide
improvements to both (enhanced) baselines: 1) applying MATE to ProtoNets+ResNet12 yields $+0.64\%$ 5-shot accuracy
and slightly better 1-shot accuracy ($+0.14\%$); 2) applying MATE to R2D2+ResNet12 yields $+0.44\%$ 5-shot accuracy
improvement and similar 1-shot accuracy ($+0.08\%$). These results are hence consistent with our original finding that
MATE brings more benefits to 5-shot accuracy than to 1-shot, which is reasonable because we can obtain more accurate
information about data distribution on the task with more data and thus task representation of higher quality.

▷ **Protocal used for meta-testing (R4).** During the meta-testing stage, we sample 1,000 episodes (Section 3.2) from
the meta-testing set following either 5-way 1-shot or 5-way 5-shot settings. The query set in each meta-testing episode
contains 15 query images over which we calculate the meta-testing accuracy. We then report the average accuracy and
standard devation of the accuracies over the 1,000 meta-testing episodes. Due to large amount of tesing episodes used,
the standard deviation of the accuracies is sometimes very close. We confirm that the numbers reported in the tables in
this paper are all correct. We would like to thank R4 for pointing out the ambiguity of the testing protocol. We will
clarify this and add more details of the experiments in the final version.

▷ **Limited performance gain (R4, R5).** Firstly, we thank R4 for the appreciation of the improvement brought by
MATE, which we think can be further strengthened by the additional experiments we just conducted. Secondly, we
humbly clarify that we calculate the accuracy standard deviation over 1,000 meta-testing tasks instead of the confidence
interval. Hence, the accuracy improvement over $0.5\%$ can show consistent improvement over a large sample of tasks.
We'd also like to emphasize that incorporating model information into task embedding does help with and improve the
performance, which is supported by the comparison of FiLM+KME and FiLM+SVM in Table 3 (2nd and 3rd rows).

▷ **Conditioning FiLM layers on $\omega$ (R5)?** If we understand correctly, R5 suggests to condition FiLM layers on the
optimal parameters learned by SVM (Section 3.1), instead of the model-aware task features proposed in this paper. We
think this question can be answered well by humbly reminding R5 of the connection of our proposed method with kernel
mean embedding (KME) [28], as we described before Section 3.1. In Eq. (1), if we ignore the model information by
taking $f_M(x) \equiv 1$, Eq. (1) reduces to KME. Further, if $\phi$ corresponds to the canonical feature map of the characteristic
kernel, the map defined by Eq. (1) is injective, i.e., the representation $\Phi(T)$ captures all information about the task $T$
[10, Lemma 2]. Therefore, condtioning FiLM layers on the model-aware task feature defined in Eq. (1), which is very
likely to contain most of information on the task, could possibly make the FiLM easier to train and, more importantly,
more interpretable. We plan to conduct comparision experiments and report related results in final version.

[Meta-Review · NeurIPS 2020]

This paper focuses on improving task embedding for meta/few-shot learning. The proposal is a model-aware task embedding that can take the task difficulty into account. The philosophy behind sounds quite interesting to me, namely, the model trained for each task is itself a key property of the task when considering transferring knowledge between different tasks. This philosophy leads to a novel algorithm design I have never seen. The clarity and novelty are clearly above the bar of NeurIPS. While the reviewers had some concerns on the significance, the authors did a particularly good job in their rebuttal. Thus, all of us have agreed to accept this paper for publication! Please include the additional experimental results in the next version.